# Current Situation of Goose Astrovirus in China: A Review

**DOI:** 10.3390/v17010084

**Published:** 2025-01-10

**Authors:** Dan Ren, Hongliang Zhang, Xiaoou Ye, Xiuzhi Jia, Ruiming Chen, Tingbing Tang, Jianqiang Ye, Songquan Wu

**Affiliations:** 1Center of Disease Immunity and Intervention, College of Medicine, Lishui University, Lishui 323000, China; 18767827825@163.com (D.R.); zhl@lsu.edu.cn (H.Z.);; 2Key Laboratory of Jiangsu Preventive Veterinary Medicine, Key Laboratory for Avian Preventive Medicine, Ministry of Education, College of Veterinary Medicine, Yangzhou University, Yangzhou 225009, China

**Keywords:** goose astrovirus, genomic diversity, pathogenesis, diagnosis

## Abstract

Gosling gout disease is an infectious disease caused by goose astrovirus (GAstV), which can result in urate deposition in the internal organs and joints of goslings. Since 2015, outbreaks of gosling gout disease have occurred in several goose-producing areas in China. Subsequently, the disease spread to the vast majority of eastern China, becoming a major threat to goose farms and causing huge economic losses to the goose industry. Meanwhile, GAstV can infect species of birds other than geese. It is worth noting that, as an emerging virus, the research on GAstV is still in the early stages. Therefore, the investigation of GAstV has become an urgent issue, which can improve understanding of GAstV and develop effective measures to control its threat to poultry. The purpose of this review is to summarize the latest research progress on GAstV in recent years, mainly focusing on the genetic evolution, pathogenesis, diagnostic detection, and control strategies of GAstV, aiming to provide a reference for scientific prevention and control of GAstV infection.

## 1. Introduction

Astroviruses (AstVs) belong to *Astroviridae*, which are non-enveloped, positive-sense, single-stranded RNA viruses showing icosahedral shapes with a diameter of 28–30 nm [1,2,3,4,5]. The viruses were first discovered in 1975 in the feces of children with vomiting and mild diarrhea. Upon electron microscopy observation, its surface displayed a distinctive star-like structure, hence named astrovirus [6,7,8]. AstVs are currently classified into two genera, *Mamastroviruses* (*MAstVs*) and *Avastroviruses* (*AAstVs*), whose members are known to contain 19 and 3 types, respectively [4,9]. In addition, many other MAstVs and AAstVs have been gradually discovered, including passerine birds, dromedary camels, shrews, pikas, and geese [10,11,12,13,14].

Since 2015, a gosling gout disease has been endemic in China and has spread to several provinces, including Guangdong, Shandong, Jiangsu, Anhui, and Fujian Provinces (Figure 1) [11,15,16,17,18,19]. It mainly affected goslings that were 1–20 days old, characterized by gout in internal organs and joints, as well as kidney swelling and bleeding (Figure 2). The lower the age, the higher the morbidity and mortality of the goslings [17,19]. It is reported that the incidence rate in goslings in some provinces is as high as 86.1%, and the mortality rate is more than 80% [17,20,21]. With further research, researchers isolated a new astrovirus of goose origin from the tissues of goslings with gout [11,19,22,23]. Meanwhile, Zhang et al. conducted animal regression experiments and successfully confirmed that goose astrovirus (GAstV) can replicate gosling gout diseases [11]. An increasing amount of research has indicated that GAstV may be the causative pathogen of gout in goslings [15,16,17,24,25]. In the early stages of infection, goslings exhibit signs of depression, sluggish movement, and reduced feed intake. As the disease progresses, they become anorectic, reluctant to walk, and may even become paralyzed. Eventually, the geese died due to excessive exhaustion, and the duration of the disease was 3–7 d. Some geese did not die but suffered a serious decline in production performance in the following time, with growth hinderance, slow development, and decreased immunity and resistance, and were more prone to other secondary bacterial or viral infections [26,27]. GAstV has a wide range of tissue tropism and can cause lesions in multiple tissues and organs of sick geese. Uric acid salt distribution can be seen in the liver, heart surface, ureters, and joint cavities of goslings infected with GAstV; the kidneys are swollen and pale, and the spleen is swollen and necrotic [28,29,30]. In addition to causing gout, GAstV infection leads to encephalitis, in which microglia in the cerebral cortex markedly proliferate and encapsulate dying neurons [17,19]. The villi of the cecum of goslings infected with GAstV are markedly shortened, which may lead to inhibition of the barrier function of the mucosa [31]. Notably, GAstV infection can cause a decrease in resistance, which can lead to other secondary bacterial or viral infections [26,27,32,33]. Moreover, in recent years, there has been abundant evidence that GAstV can cross the species barrier and infect other birds, such as muscovy ducks and chickens, which has increased public concern about GAstV [34,35,36,37]. This article reviews the current research state of GAstV in poultry by discussing the genetic composition and diversity, pathogenesis, diagnosis, and current strategies in the control of GAstV.

## 2. Genome Organization and Genomic Diversity of GAstV

Astroviruses are single-stranded positive-stranded RNA viruses, and the viral particles are visible under electron microscopy as a star-shaped structure with a diameter of 28–30 nm (Figure 3) [2,38]. The genome length of GAstV is approximately 7.2 kb, which includes a 5′-untranslated region (UTR), three open reading frames (ORFs) (ORF1a, ORF1b, and ORF2), a 3′-UTR, and a poly(A) tail [16,39]. Among them, ORF1a and ORF1b encode nonstructural proteins (NSPs) involved in RNA transcription and replication, while ORF2 encodes the structural proteins, which are expressed from a subgenomic RNA and known as capsid (Cap) proteins [40,41]. ORF1a encodes a putative helicase domain (HEL), several transmembrane (TM), a coiled-coil (CC) region, a protease domain (PRO), a zinc finger-like motif (ZNF), a nuclear localization signal (NLS), as well as several other proteins of unknown function (Figure 4) [42]. ORF1b encodes the RNA-dependent RNA polymerase (RdRp). There is an overlapping region between ORF1a and ORF1b, which was a highly conserved seven-ribosome frameshift sequence (5′-AAAAAAC-3′) and was regarded as a ribosome translocation signal (RFS) [42,43,44]. ORF2 contains the region coding for the shell (S) domain, a P1 domain of unknown function, and a P2 domain that includes the spike protein [3]. Among them, S and P1 domains are conservative regions, while P2 is a highly variable region. Spike is the structural barrier of the astrovirus, involved in recognizing cell surface-related receptors and host immune response, and is also an important area mediating invasion [45,46].

Based on the differences in the amino acid sequences of the GAstV ORF2, GAstV can be categorized into two serotypes, GAstV-1 (e.g., GAstV-FLX strain) and GAstV-2 (e.g., GAstV-GD strain) [47]. However, due to conflicting accounts, categorization and harmonization of subsequent studies are awaited. By sequence alignment, the genome-wide similarity of GAstV-1 and GAstV-2 was 57.9%, and the ORF2 genome similarity was 42.6% [30]. The amino acid sequence of the coat protein of the GAstV-1 strain is in the same branch as that of chicken astrovirus (CAstV); GAstV-2 is closer to turkey astrovirus (TAstV) (Figure 5). It is worth noting that all strains in GAstV-2 are related to the outbreak of gout in goslings, and the strain under the classification of GAstV-1 is mainly characterized by gosling enteritis when it was first found, but in subsequent studies, it also began to show goose gout similar to GastV-2 [22,47,48]. However, the incidence rate and mortality of GastV-2 are far higher than GastV-1 [22]. ORF2 structure prediction showed that the structures of GAstV-1 and GAstV-2 are basically similar. In only the ORF1a protein, compared to the four transmembrane regions in GAstV-2, the transmembrane region in GAstV-1 increased by one (Figure 4). Various studies have shown that transmembrane play important roles in the endocytosis or cell fusions of the virus [49,50,51], which could help in driving and anchoring the nonstructural replication complexes on cellular membranes [6,52].

Similar to most RNA viruses, the AstV RdRP is error-prone and lacks proofreading capabilities, resulting in susceptibility to nucleotide mutations and genomic reorganization during replication [53,54]. A higher genetic variability in ORF2 than in ORF1a and ORF1b has been confirmed. Mutations and/or recombination of ORF2 have the potential to alter the ability of the virus to attach to and enter the host cell. This may help the virus to recognize new cellular receptors or adapt to those of a new species, thus facilitating its cross-species transmission. Compared to GAstV-1, GAstV-2 is currently widespread in China. To gain information on the evolution of amino acid residues on ORF2 from GAstV-2 over time in nature, we analyzed ORF2 proteins of GAstV-2 isolates submitted to NCBI from 2018 to 2024. The results indicate that nine mutated amino acid residues in the ORF2 protein, including Y36H/S, V60I, A228T/S/V, Q229P, A289T, T376A, E456N/D, L540Q, and A614N/T (Figure 6). Among these, the amino acid variations observed in 2021 were particularly significant, with a notably high proportion of transitions between A and T at these sites. However, the precise impact of alterations at these loci on GAstV-2’s infectivity and transmission dynamics remains unclear. It is worth noting that a study has established a successful reverse genetics technique for the development of GAstV-2/JS2019 strain [55]. This may be an important contribution to monitoring the evolution and spread of GAstV-2 and to the investigation of the biology of the newly discovered virus.

## 3. Pathogenesis of GAstV

The pathogenesis of GAstV-induced gout in goslings is mainly due to excessive production of uric acid, decreased excretion of uric acid, and renal injury [56]. Due to the lack of arginase in poultry, the ammonia produced by metabolism cannot be converted to urea and can only be excreted in the form of uric acid through the purine nucleotide synthesis and decomposition pathway [57,58]. Because uric acid is difficult to dissolve in water, it easily reacts with calcium and sodium to form calcium and sodium uric acid, which are deposited on the surface of the viscera, renal tubules, and articular cavities.

Previous studies have shown that GAstV infection significantly increased serum uric acid, creatinine, alanine aminotransferase (ALT), aspartate aminotransferase (AST), and γ-glutamyltransferase (γ-GT) levels and reduced body weight [28,30,31,59,60]. On the other hand, the activity and expression levels of xanthine dehydrogenase (XOD) and adenosine deaminase (ADA) in the liver of goslings infected with GAstV were significantly higher than those in the same tissues of the control group. Meanwhile, as the main transporters of uric acid in poultry, multidrug resistance protein 4 (MRP4) mRNA expression and Na-K-ATPase activity were significantly reduced, leading to decreased renal excretion function [30,59,61,62]. Meanwhile, GAstV infection caused renal epithelial cell autophagy, destruction of brush borders and intercellular junctions, podocyte damage, and increased fibrosis, ultimately resulting in damage to the kidney [30]. These results all showed that infection with GAstV could cause liver and kidney damage, leading to enzyme system disorders, excessive production of uric acid, and decreased excretion of uric acid, which can lead to the formation of hyperuricemia and gout [59,63]. Moreover, GAstV infection could also lead to necrosis of reticular cells, destruction of reticulocyte fibers, apoptosis of lymphocytes, and decreased levels of CD8, resulting in splenic injury [29]. The global transcriptome and metabolic network pathways in the kidneys of goslings infected with GAstV showed that as GAstV replication increased in vivo, the regulation of key enzymes in the host metabolism progressively increased, flowing metabolites into the purine/pyrimidine biosynthesis pathways [64]. GAstV inhibited the host oxidation-reduction response by inhibiting the expression of the catalase gene and activated the key enzyme in lactic acid synthesis to produce lactate accumulation, which inhibits the host’s antiviral response. RNA-seq identification of differentially expressed genes after GAstV infection revealed partial downregulation of metabolism-related pathways, such as tryptophan metabolism, drug metabolism by cytochrome P450, and drug metabolism by other enzymes and peroxisome [65]. In contrast, pathways associated with host cell defense and proliferation appeared to be upregulated, including extracellular matrix receptor interactions, complement and coagulation cascades, the PI3K-Akt signaling pathway, lysosomes, and the tumor necrosis factor signaling pathway. In addition, GAstV infection altered the gut microbiota of goslings, with the enrichment of potential pro-inflammatory bacteria and depletion of beneficial bacteria that can produce short-chain fatty acids [66]. Notably, the microbial pathway involved in urate production was significantly increased in goslings infected with GAstV, suggesting that gut microbiome-derived urate may also contribute to the gout symptoms.

Activation of the NOD-, LRR- and pyrin domain-containing protein 3 (NLRP3) inflammasome and release of Interleukin-1beta (IL-1β) play critical roles in the inflammatory response in gout [67]. Urate crystals interact with macrophages and activate NLRP3, which recruits and activates Caspase-1. The catalytic effect of Caspase-1 can promote the conversion of the inactive precursors pro-IL-1β and pro-IL-18 into mature, bioactive IL-1β and IL-18, thereby inducing severe acute inflammatory reactions in the body [68]. Research has shown that GAstV infection results in high expression of IL-1β and IL-8 in the spleen, as well as high expression of inducible nitric oxide (NO) synthase (iNOS) in the spleen and kidneys [69]. In addition, GAstV infection induced the activation of pattern recognition receptors (Retinoic acid-inducible gene-I (RIG-I), Melanoma differentiation-associated gene 5 (MDA-5), Toll-like receptors 3 (TLR3)) and key junction molecules (Myeloid differentiation primary response protein 88 (MyD88), Mitochondrial antiviral signaling protein (MAVS), Interferon regulatory factor 7 (IRF7)) in the spleen and kidney and upregulated the gene expression of IL-6 and interferon-α and antiviral proteins (MX dynamin like GTPase 1 (MX1), 2′-5′-Oligoadenylate synthetase like (OASL), Interferon induced transmembrane protein 3 (IFITM3)) [28,39,69]. These results indicate that GAstV infection activates the host’s innate immune response. Meanwhile, GAstV infection increased the expression levels of CD81, Major histocompatibility complex (MHC) I, and MHCII, indicating that adaptive immune responses were activated [69]. In addition, GAstV infection activated the TLR2 pathway and enhanced the immune inflammatory response [64]. Inflammatory damage to the kidneys further exacerbated renal excretory dysfunction [64,69].

## 4. Diagnosis of GAstV

Electron microscopy is one of the earliest methods used to detect GAstV by observing its characteristic five- or six-pointed star-like structure with a diameter of 28–30 nm [3,4,5]. Yuan et al. found virus particles of approximately 30 nm in diameter in infected liver tissue through a transmission electron microscope [70]. At the same time, Zhu et al. found spherical virus particles with a diameter of 28 nm after negative staining with phosphotungstic acid under electron microscopy [23].

The isolation and identification of viruses have always been the gold standard for detecting infectious diseases. Inoculating goose embryos is currently a commonly used method for isolating GAstV. GAstV can be amplified and cultured by inoculating 9–13-day-old goose embryos via yolk sac or chorioallantoic membrane [44,66,71,72,73]. At the same time, the thickening of the chorioallantoic membrane and different levels of hemorrhage were observed in the dead goose embryos. It is worth noting that the study has found that GAstV only reproduced in goose embryos, not in SPF chicken or duck embryos [70]. However, due to the difficulty in obtaining goose embryos and the lack of SPF goose embryos, it is difficult to isolate and identify GAstV. Therefore, it is significant to find a suitable cell system for GAstV expansion culture for virus isolation. In 2018, Zhang et al. first discovered that GAstV could be isolated and cultured using LMH cells, a chicken liver cell line, with the addition of appropriate trypsin-tosylsulfonyl phenylalanyl chloromethyl ketone (TPCK) [11]. Subsequently, other researchers confirmed this finding [74,75,76]. Meanwhile, some scholars have used goose embryo kidney cells (GEK cells), goose kidney tubular epithelial cells (GRTE cells), chicken embryo kidney cells (CEK cells), duck embryo liver cells (DEH cells), and duck embryo kidney cells (DEK cells) to isolate and culture GAstV [35,65,77]. However, as primary cells, the preparation process of these cells is relatively complicated and not an ideal virus in vitro proliferation system.

Serological detection methods are also important detection methods, such as indirect immunofluorescence assays (IFAs), enzyme-linked immunosorbent assays (ELISAs), etc., which can be used to detect GAstV. In 2018, Zhang et al. detected GAstV in LMH cells through IFA using the convalescent sera from the survival geese [11]. In addition to the use of polyclonal antibodies (pAbs), monoclonal antibodies (mAbs) are increasingly being considered as a promising diagnostic agent for determining viral titers due to their ability to produce large amounts of specific antibodies [78,79]. A mAb-based sandwich ELISA for the detection of GAstV was developed using mAb 6C6 against ORF2 protein. Moreover, the tissue-frozen section-IFA approaches for efficient detection of GAstV were developed [5]. In this context, one novel peptide of 627–646 aa in the ORF2 recognized by mAb 6C6 was used as an antigen to develop an efficient peptide-based ELISA (pELISA) for detection of antibodies against GAstV [80]. In addition, a highly sensitive indirect competitive ELISA (ic-ELISA) based on a mAb was developed to detect GAstV-specific antibodies from geese. A novel diagnostic immunochromatographic strip (ICS) assay based on antibody colloidal gold nanoparticles specific to GAstV was developed to detect GAstV in goose allantoic fluid and supernatant of tissue homogenate. The lower limit of the ICS was reported to be approximately 1.2 μg/mL [81].

The reverse transcription-quantitative PCR (RT-qPCR) method is currently the most widely used method in laboratories and has been developed for rapid detection of GAstV [82]. TaqMan-based quantitative RT-PCR has also been employed to detect and quantify GAstV, which can be used to detect GAstV in both field samples, embryos, and newly hatched goslings [82,83]. Due to the complex co-infections of GAstV, different genotypes and different pathogens always occur, and the duplex or multiplex PCR detection method has been established [84,85,86,87,88,89,90]. In addition, a droplet digital polymerase chain reaction (ddPCR) system has been developed to sensitively and accurately quantify GAstV using the conserved region of the ORF2 gene, and this detection limit of ddPCR was 10 copies/μL [91]. An isothermal detection method based on reverse transcription loop-mediated isothermal amplification (RT-LAMP) was established for rapid and easy detection of GAstV [92,93]. The detection limit of the RT-LAMP assay was 50–100 copies/μL, and the sensitivity was tenfold higher than that of the conventional RT-PCR. Meanwhile, a novel diagnostic test for detecting GAstV was developed by combining reverse transcription-enzymatic recombinase amplification (RT-ERA) and Clustered regularly interspaced short palindromic repeats (CRISPR)-Cas12a technologies. This method was achieved with no cross-reaction with non-GAstV templates and a sensitivity detection limit of two copies/μL [94]. In addition, metagenomics sequencing technology was also used as a powerful tool for the detection of various pathogens, including GAstV [48,95,96].

## 5. Treatment and Control of GAstV

Due to the widespread occurrence of GAstV in goose farms, coupled with their environmental stability and resistance to inactivation by most disinfectants, it has become difficult to eliminate GAstV from affected areas [24]. Notably, reports showed that GAstV RNA was detected in the yolk membranes, embryos, and allantoic fluid of embryos laid by geese inoculated with GAstV and had almost 100% nucleotide homology with the virus isolated from the ovaries of geese. These results indicate that GAstV can be transmitted vertically in geese, which poses a great obstacle to the prevention and control of GAstV [74]. Therefore, the prevention of GAstV infections is essentially based on the control of the transmission routes and on the prevention and control of disease at the host level.

Strict biosafety measures, extended downtime between production cycles, and the use of effective disinfectants may reduce the likelihood of GAstV infection [97,98]. GAstV is mainly transmitted through the digestive tract, so goose farms should collect and clear poultry feces regularly and carry out disinfection work on the environment and facilities strictly. Studies have shown that disinfecting with 1 mg/mL of free chlorine for 2 h is quite effective against AstV, but there may be differences in environmental persistence among different strains [99]. Meanwhile, it is important to extend the idle time between different animal batches, improve feeding methods, and reduce feeding density [3,100]. Avoid contact between sick geese and other healthy goslings, and strictly handle dead animals in accordance with the regulations for harmless treatment. High protein, high calcium levels, low phosphorus levels, and vitamin A deficiency in feeds can lead to gout [101,102,103]. Therefore, the use of feeds containing high protein and purine should be controlled. At the same time, drugs for protecting the liver and kidneys can be used to improve liver detoxification and kidney metabolism. Studies have shown that NO and iNOS levels were significantly higher in the kidneys and spleens of GAstV-infected goslings [29,30]. Significantly, aminoguanidine (AG), an iNOS inhibitor, can significantly reduce the serum NO concentration and alleviate kidney disease in goslings infected with GAstV. Moreover, AG reduces mortality, serum uric acid, and creatinine content, decreases uric acid deposition in internal organs and joints, and alleviates renal tubular cell necrosis and inflammatory cell infiltration [77]. In addition, preventing kidney damage is important for reducing the incidence of gout in goslings [104]. Currently, there is no commercial veterinary drug for gout. Therefore, kidney-protecting diuretics can be used to promote the excretion of uric acid when gout occurs. For more serious conditions, comprehensive treatment measures such as fluid replacement and support can be taken to prevent dehydration or other serious complications.

Vaccine research is also an important direction for preventing GAstV. Some studies have shown that the specific IgY produced by inactivated GAstV immunization can effectively protect goslings. The results of prevention and treatment experiments indicated that the prevention and cure rates exceed 80% when yolk antibodies are administered in the early stages of GAstV infection [75]. It is worth noting that, clinically, there are mixed infections between various subtypes of GAstV and other viruses, such as goose parvovirus (GPV) and goose circovirus (GCV), which increase the severity of gout [26,27,32,33]. Therefore, multivalent vaccines based on avian adenovirus, as well as other avian-derived viruses as vaccine vectors, deserve further research. In a study, the modified goose-origin Newcastle disease virus (NDV) recombinant vaccine virus expressing the Cap protein of GAstV protected against both GAstV and avelogenic NDV infections and could potentially become a safe, stable, and effective bivalent vaccine [105]. Unfortunately, although there have been reports of experimental vaccines, there is currently no commercially available vaccine for preventing GAstV infection [19,47].

## 6. Conclusions

Goose astrovirus mainly causes gout in goslings and has high morbidity and mortality. It has caused large-scale epidemics in China over the past few years, resulting in serious economic losses to the goose industry. GAstV is transmitted both vertically and horizontally [74]. At the same time, due to its ability to mutate and recombine, GAstV has shown cross-species transmission [34,36]. The field of GAstV research has undergone rapid expansion; however, most are limited to epidemiological investigations and genetic diversity analysis. The understanding of its infection and mechanism is still limited and needs to be further explored. Moreover, in prevention, good management systems are by far the best method of preventing and controlling GAstV. There is no commercialized vaccine for GAstV, but with a deeper understanding of the molecular biology characteristics of GAstV, the vaccines, such as DNA vaccines and recombinant vaccines, are a good area of research. There is still a long way to go to study and prevent GAstV.

## Figures and Tables

**Figure 1 viruses-17-00084-f001:**
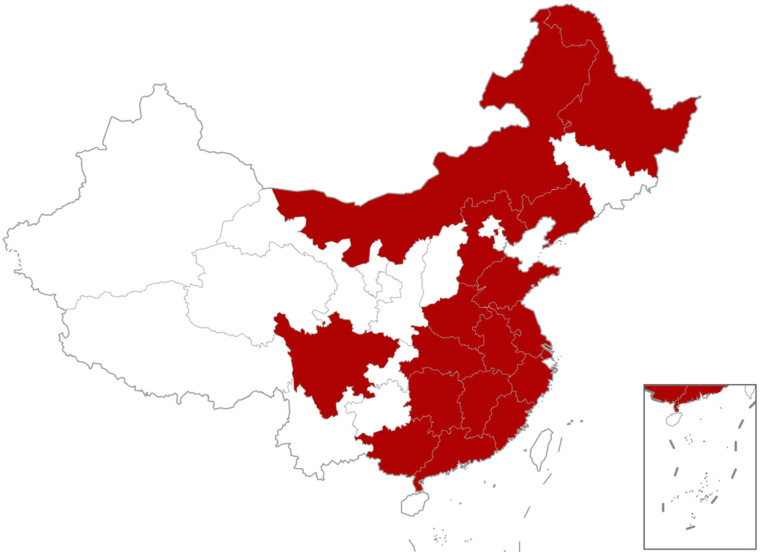
Regions of goose astrovirus (GAstV) infection in China. The provinces reporting GAstV infection were indicated in red.

**Figure 2 viruses-17-00084-f002:**
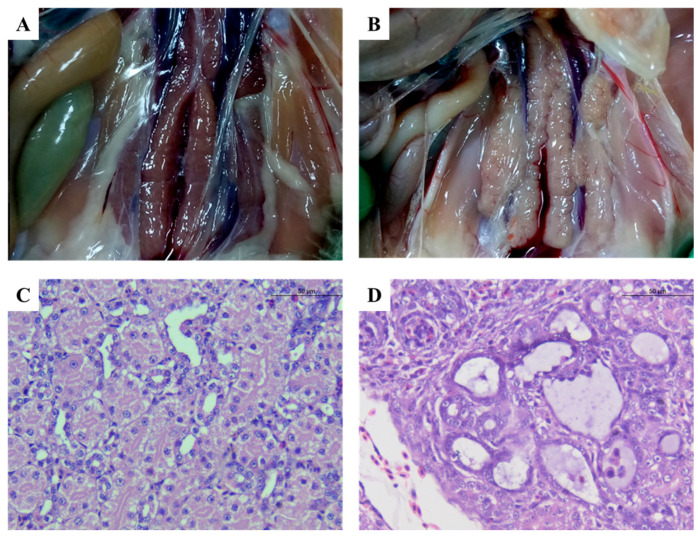
Gross and microscopic pathologic changes in the kidneys of diseased goslings. (**A**,**C**) Normal kidney from the non-infected gosling. (**B**,**D**) Kidney from the infected gosling; pale and swollen, with severe dilation of renal tubules and protein cast in renal tubules.

**Figure 3 viruses-17-00084-f003:**
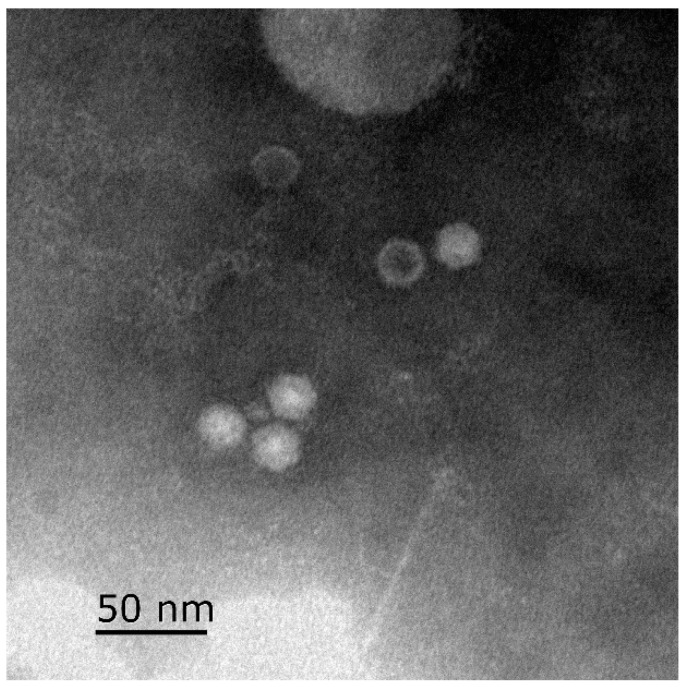
GAstV virus particles under the electron microscope. Bar: 50 nm.

**Figure 4 viruses-17-00084-f004:**
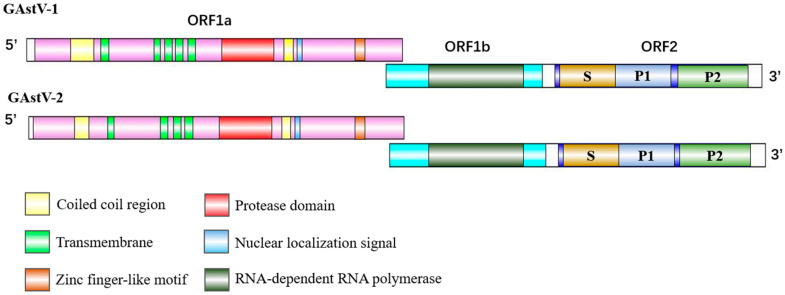
The genome structure of GAstV-1 and GAstV-2.

**Figure 5 viruses-17-00084-f005:**
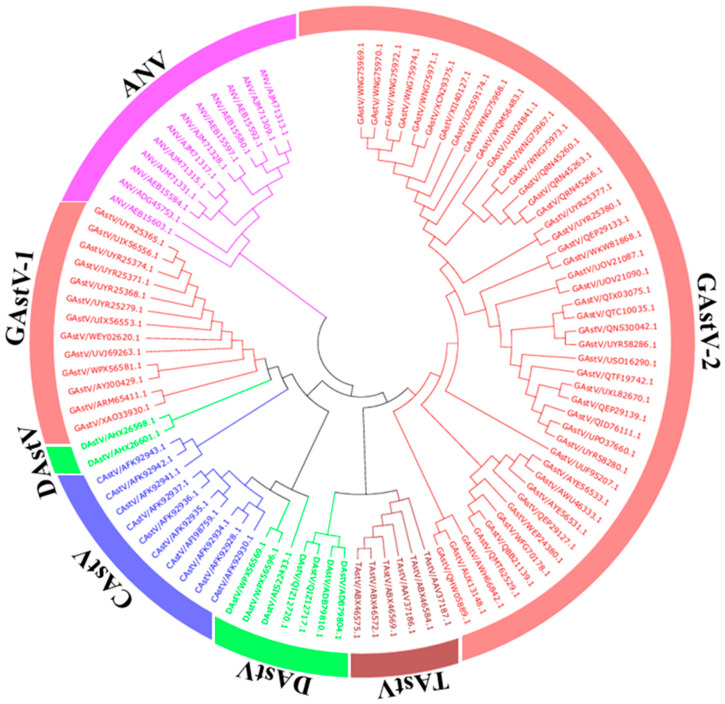
Phylogenetic analysis of the complete *Avastroviruses* (*AAstVs*) sequence using MEGA 7.0. Each background color represents an astrovirus species.

**Figure 6 viruses-17-00084-f006:**
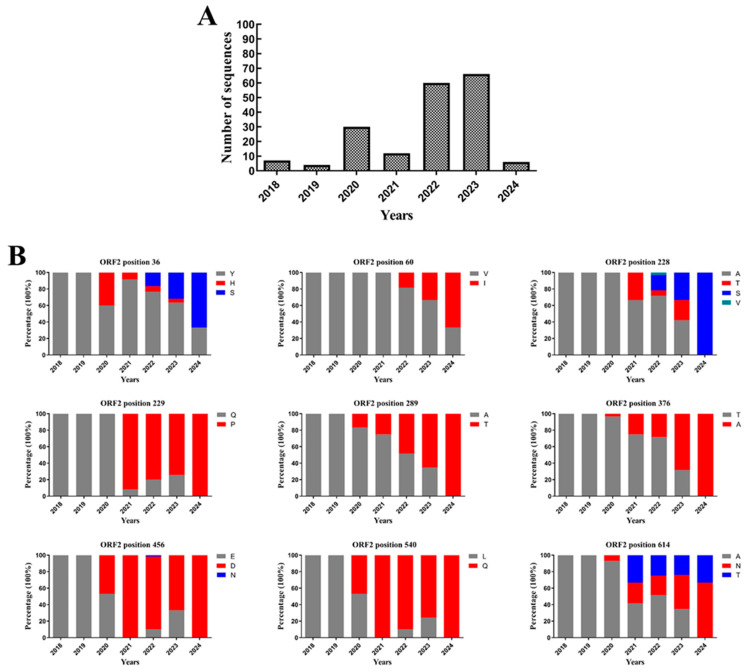
Amino acid mutations on open reading frame 2 (ORF2) of GAstV-2 isolated strains. (**A**) The number of ORF2 sequences analyzed by years. (**B**) The percentage of amino acid mutated on ORF2.

## Data Availability

Not applicable.

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
