# Peer review of "Current Situation of Goose Astrovirus in China: A Review"

_viruses, 2025, doi:10.3390/v17010084_

Round 1
Reviewer 1 Report
Comments and Suggestions for Authors
Review by Ren et al. entitled “Current Situation of Goose Astrovirus in China: A Review” (viruses-3359741) described recent trend in the research field on Goose astrovirus and its epidemics. Their manuscript was easy to read, nicely balanced, and concise yet detailed enough to provide organized view on the Goose astrovirus infection and its pathogenesis with emphasis on its diagnosis and disease control. This reviewer has no objection to publish this manuscript in Viruses.
Followings are the minor points that may require authors’ attention.
1. It may be helpful for readers to present some hard data on the economical and agricultural impact by Goose astroviral epidemics, such as estimated amount of loss or proportion/number of gosling damaged by the virus.
2. In Fig.1, why Guangxi was not included as GAstV affected area? Previous review by others was, instead (Biomed Res Int. 2022 Aug 28:2022:1635373.).
Author Response
审稿人 #1:任 等人的评论题为“中国鹅星状病毒的现状:综述”(viruses-3359741) 描述了鹅星状病毒及其流行病研究领域的最新趋势。他们的手稿易于阅读,平衡良好,简洁而又足够详细,以提供关于 Goose 星状病毒感染及其发病机制的有条理的观点,重点是其诊断和疾病控制。本审稿人不反对在 Viruses 上发表此手稿。
以下是可能需要作者注意的次要要点。
- 提供一些关于 Goose 星状病毒流行病对经济和农业影响的硬数据可能会对读者有所帮助,例如估计的损失量或被病毒破坏的小鹅的比例/数量。
谢谢你指出这一点。我们在第 37-39 行添加了相关内容。
- 在图 1 中,为什么广西没有被列为 GAstV 影响地区?其他人之前的评论是 (Biomed Res Int. 2022 Aug 28:2022:1635373.)。
我们真的很抱歉我们粗心大意的错误。感谢您的提醒。我们对图 1 进行了修改。

Reviewer 2 Report
Comments and Suggestions for Authors
The presented review is of utmost importance to the economic perspective, as it has tried to address one of the unexplored viruses of the host belonging to the Anatidae family.
While going through the manuscript, I found some important deficiencies, which need to be addressed before it gets published.
1. The review lacks considerable information about the disease transmission and distinct clinical symptoms of the birds with relevant figures.
2. There were no pathological gross lesions/ microscopic pathologic figures to show the distinct tissue changes due to exclusive GAstV infections.
3. The manuscript didn't touch upon on the immunologic aspects of the disease in geese (in detail).
4. While discussing epidemiology, the author should have touched upon the distribution of diseases in some other parts of the world to inform readers about the actual impact of the diseases at the global level.
5. A sub-section of differential diagnosis is also needed to allow the reader to know about GAstV infection and how it differs from other similar diseases of goose/ poultry.
Besides this, there are several points that authors are requested to address:
1. Lines 84,85 and 86: there is an imperceivable distinction between the pathologic effect discussed, between GAstV 1 and GAstV2. Needs a little more elaboration.
2. Line 89: transmembrane ?? not clear, some additional lines/text is missing.
3. Line 91: I wonder, how nonstructural protein could help in transmembrane fusion, in place of structural protein? Please check.
4. Line 108: Give reference to this work.
5. Line 135: Give reference to this work.
6. Line 145: Reticulocyte fibers are not clear?? Please check.
7. Line 219: A suitable reference is missing here.
8. Line 291: The authors concluded the text on vertical transmission without giving much information about the actual transmission of the diseases in any form in the text. A more detailed and elaborative response is needed in each sub-section requested above.
Author Response
Dear Editors and Reviewers:
Thank you for your letter and for the reviewers’ comments concerning our manuscript entitled “Current Situation of Goose Astrovirus in China: A Review” (ID: viruses-3359741). Those comments are all valuable and very helpful for revising and improving our paper, as well as the important guiding significance to our researches. We have studied comments carefully and have made correction which we hope meet with approval.
We sincerely thank the editor and all reviewers for their valuable feedback that we have used to improve the quality of our manuscript. The reviewer comments are laid out below in italicized font and specific concerns have been numbered. Our response is given in normal font and changes/additions to the manuscript are given in the red text.:
Responds to the reviewer’s comments:
Reviewer #2: The presented review is of utmost importance to the economic perspective, as it has tried to address one of the unexplored viruses of the host belonging to the Anatidae family. While going through the manuscript, I found some important deficiencies, which need to be addressed before it gets published.
- The review lacks considerable information about the disease transmission and distinct clinical symptoms of the birds with relevant figures.
Thank you for pointing this out. We have added related content on line 44-54.
- There were no pathological gross lesions/ microscopic pathologic figures to show the distinct tissue changes due to exclusive GAstV infections.
Thank you for pointing this out. We have added related content in Figure 2.
- The manuscript didn't touch upon on the immunologic aspects of the disease in geese (in detail).
Thank you for pointing this out. There has not been in-depth research on goose astrovirus in this area. And The "pathogenesis of GAstV" has been confirmed to mention relevant content, hence we have not provided a detailed description of this part.
- While discussing epidemiology, the author should have touched upon the distribution of diseases in some other parts of the world to inform readers about the actual impact of the diseases at the global level.
Thank you for the suggestion. At present, the outbreak of goose astrovirus is only in China, and there are no relevant reports abroad, hence we did not mention it in the review.
- A sub-section of differential diagnosis is also needed to allow the reader to know about GAstV infection and how it differs from other similar diseases of goose/ poultry.
Thank you for pointing this out. At present, gout caused by GAstV infection is a very typical symptom, so we did not mention the differential diagnosis between avian influenza virus and goose disease in this article.
Besides this, there are several points that authors are requested to address:
- Lines 84,85 and 86: there is an imperceivable distinction between the pathologic effect discussed, between GAstV 1 and GAstV2. Needs a little more elaboration.
Thank you for the suggestion. We have modified related content on line 101-104.
- Line 89: transmembrane ?? not clear, some additional lines/text is missing.
Thank you for pointing this out. We have modified related content on line 107-111.
- Line 91: I wonder, how nonstructural protein could help in transmembrane fusion, in place of structural protein? Please check.
Thank you for raising this issue. This section was revised and modified.
- Line 108: Give reference to this work.
Thank you for the suggestion. We have added relevant reference.
- Line 135: Give reference to this work.
Thank you for the suggestion. We have added relevant reference.
- Line 145: Reticulocyte fibers are not clear?? Please check.
Thank you for raising this issue. This section was revised and modified.
- Line 219: A suitable reference is missing here.
Thank you for the suggestion. We have added relevant reference.
- Line 291: The authors concluded the text on vertical transmission without giving much information about the actual transmission of the diseases in any form in the text.
Thank you for pointing this out. We have modified related content on line 279-284.
We would like to thank you once again for your valuable time and suggestions. We are confident that our manuscript has been significantly improved and will contribute to the research field of goose astrovirus after these revisions.
We look forward to your further feedback.
Yours sincerely,
Dan Ren

Round 2
Reviewer 2 Report
Comments and Suggestions for Authors
Minor revision needed: A better picture (microphotograph) of normal kidney can be added. There is more intense eosinophilia of the kidney tubules noted, probably due to use of poor buffer/over /prolonged eosin staining? .
Shall appreciate, if a better picture with good staining is replaced over the current one.
Rest is okay.
Thanks
Author Response
-
1.A better picture (microphotograph) of normal kidney can be added. There is more intense eosinophilia of the kidney tubules noted, probably due to use of poor buffer/over /prolonged eosin staining?
Thank you for pointing this out. We have modified the histopathological images of both normal and diseased gosling kidneys, as they were processed together.